# Microstructure and Mechanical Properties of Ni-20Cr-Eu_2_O_3_ Composites Prepared by Vacuum Hot Pressing

**DOI:** 10.3390/ma16041473

**Published:** 2023-02-09

**Authors:** Yihong Zhou, Huifang Yue, Zhaohui Ma, Zhancheng Guo, Jiandong Zhang, Lijun Wang, Guoqing Yan

**Affiliations:** 1National Engineering Research Center for Environment-Friendly Metallurgy in Producing Premium Non-Ferrous Metals, GRINM Group Corp., Ltd., Beijing 101407, China; 2State Key Laboratory of Advanced Metallurgy, University of Science and Technology Beijing, Beijing 100083, China; 3GRINM Resources and Environment Tech. Co., Ltd., Beijing 101407, China; 4General Research Institute for Nonferrous Metal, Beijing 100088, China; 5Nuclear Power Institute of China, Science and Technology on Reactor System Design Technology Laboratory, Chengdu 610200, China

**Keywords:** Ni-based composites, control rods, europium oxide, microstructure, mechanical properties

## Abstract

Ni-20Cr-Eu_2_O_3_ composites were designed as new control rod materials and were synthesized from Ni, Cr, and Eu_2_O_3_ mixture powders via ball milling and vacuum hot pressing. During ball milling, Eu_2_O_3_ was fined, nano-crystallized, amorphized, and then dissolved into matrix. The effect of Eu_2_O_3_ content on the microstructure and mechanics was researched, and the corresponding mechanism was discussed. The relative densities, grain sizes, and microhardness increased when Eu_2_O_3_ content increased. According to the TEM observations, Eu_2_O_3_ particles showed a semi-coherent relationship with the matrix. The results of mechanical property testing showed that the ultimate tensile strength, yield strength, and elongation decreased with the Eu_2_O_3_ content increased. The maximum ultimate tensile strength, yield strength, and elongation were 741 MPa, 662 MPa, and 4%, respectively, with a 5 wt.% Eu_2_O_3_ addition. The experimental strengths were well matched with the theoretical values calculated by the strengthening mechanisms indicating that this method was highly effective for predicting the mechanical properties of Ni-20Cr-Eu_2_O_3_ composites.

## 1. Introduction

The control rod plays a very important role in nuclear reactors, which are used to compensate for fuel consumption and adjust the reaction rate. It is commonly used in reactors mainly for boron carbide (B_4_C), Ag-In-Cd alloy, hafnium (Hf), and rare earth elements. B_4_C has been widely used in nuclear reactors because of its large neutron absorption cross-section; however, large irradiation damage, such as from swelling and cracking, was induced by (n, α)-reactions [1]. Although Ag-In-Cd alloys are successfully used in commercial pressurized water reactors, the nuclear reactivity of control rod assemblies drops to about 80% of its initial value after five years of operation. Hf has been used as a neutron absorber in the advanced test reactor (ATR); however, because it is difficult to separate from zirconium, elemental Hf is a costly material [2]. Dy_2_TiO_5_ pellets are used as neutron absorbers in Russian thermal reactors, such as VVER-1000 and RBMK, but their neutron absorption capacity decreases quickly with an increase in service time [3,4,5]. Meanwhile, with developments in nuclear reactors, such as VHTR and GFR reactors in Generation IV, the operating temperatures are much higher than the reactors of the past generations [6]. Therefore, it is of great significance to develop new control rod materials with excellent comprehensive performances to be in service for a long time.

According to nuclear property assessment and burnup analysis, europium (Eu) is an excellent candidate for use as a control rod material [7,8,9]. Eu has a good absorption cross-section; moreover, the Eu isotope produced after irradiation has excellent neutron absorption capacity for a long lifetime, and it has been used in the Russian BOR-60 and BN-600 reactors [8]. However, Eu will not be used in a reactor in the metallic state since the metal is extremely reactive toward water and oxygen. Therefore, ceramics or metal matrix composites (MMCs), such as europium oxide (Eu_2_O_3_) pellets and Eu_2_O_3_-HfO_2_/ZrO_2_, are selected as control rod materials [10,11,12,13]. Ceramic materials have low thermal conductivity and produce thermal stress at high temperatures, leading to material failure. Metal-based composites are designed and considered excellent neutron absorber control rod materials.

At high temperatures, Ni-based alloys are usually expected to be a matrix material because they have higher creep resistance, especially at above 600 °C compared with austenitic steels and ferritic-martensitic steels [14,15,16,17,18]. Ni-20Cr has been used for many years and was one of the first alloys that could operate at high temperatures for long times without undergoing significant degradation [19,20,21,22,23]. Leitten et al. prepared stainless-based Eu_2_O_3_ composites with high physical efficiency and good thermophysical properties [24]. However, there has been less discussion as of yet about the microstructure and mechanical properties of Ni-20Cr-Eu_2_O_3_ composites. Hence, it is of great importance to investigate the microstructure and mechanical properties of the composites.

Thus far, the microstructure and mechanical properties of Ni-20Cr-Eu_2_O_3_ hot-pressing sintering composites have not been reported. In this study, Eu_2_O_3_ with different mass fractions was used to fabricate Ni-20Cr-Eu_2_O_3_ composites using ball milling and vacuum hot-pressing. The microstructures of ball-milled powders and sintered bulks were investigated and discussed. The phase composition, grain size distribution, crystal structure, and misorientation relationship of Ni-20Cr-Eu_2_O_3_ composites were systematically analyzed. Based on the microstructure observation, the strengthening efficacies of various different strengthening mechanisms were quantitatively calculated.

## 2. Materials and Methods

### 2.1. Preparation of Milled Powders and Consolidated Samples

The raw materials were Ni powder (particle size of 50 μm, purity of 99.99%), Cr powder (particle size of 15 μm, purity of 99.95%), and Eu_2_O_3_ powder (particle size of 5.5 μm, purity of 99.9%) and were all procured from the GRINM Group Corp. Ltd. (Beijing, China), as shown in Figure 1. The powder mixtures of Ni-20Cr and various amounts of Eu_2_O_3_ powders (based on Table 1) were mechanically alloyed by a planetary ball mill (QXQM-8, Changsha Tianchuang Powder Technology Co., Ltd., Changsha, China) using a corundum jar and zirconia balls. The milling was conducted at a speed of 400 r/min and for a period of 2 h. The powder mixtures were poured into a graphite die (86 mm in diameter and 25 mm in thickness). The consolidation of the as-milled powders was conducted by hot-press sintering (916G-G Press, Thermal Technology, Santa Rosa, CA, USA) protected by high-purity argon under the condition of a pressure of 50 MPa. As ball-milled powder was sintered at 1250°C, the heating rate was less than 10 °C/min, and the holding time was 2 h. After hot-pressing, samples with a diameter of 86 mm and a thickness of 8 mm were obtained.

### 2.2. Materials Characterization

The phase structures were determined using X-ray diffraction (XRD, Rigaku SmartLab, Tokyo, Japan) on a diffractometer equipped with Cu Kα radiation (λ = 1.5406 Å). The morphologies of the raw materials, as-milled powders and microstructures of hot-pressed samples, and fracture surfaces of tensile surfaces of tensile samples were characterized by using a field-emission scanning electron microscope (FESEM, JEOL JSM-7900F, JEOL, Tokyo, Japan). To measure the grain sizes, the electron backscattered diffraction (EBSD) technique was employed using a TSL system (EDAX/TSL Hikari EBSD, Hikari, Draper City, UT, USA) attached to the FESEM. The data were analyzed and processed using the OIM 7.0 analysis software. The microstructure, distribution of nanoparticles and interfaces of Eu_2_O_3_/matrix in composites were observed by a high-resolution transmission electron microscopy (HRTEM, FEI Talos F200X G2, Thermo Scientific™, Waltham, MA, USA), and EDS was used to analyze the composition. The TEM samples were prepared using the ion beam thinning technique.

The physical density of the samples was measured using Archimedes’ principle, with at least six measurements for each sample. The densities of Ni, Cr, and Eu_2_O_3_ used for theoretical density calculations were 8.90 g/cm^3^, 7.20 g/cm^3^, and 7.42 g/cm^3^, respectively. The microhardness of the samples was performed by a Wilson Hardness Tukon 250 tester (Wilson® Hardness, Binghamton, NY, USA), with a test load of 1000 g and a pressure holding time of 15 s. The mechanical properties, including yield strength, tensile strength, and elongation, were measured with a universal material testing machine (Shimadzu AG-250KNIS, Shimadzu Corporation, Tokyo, Japan). The samples were cut into dog bone-shaped specimens of 35 mm in gauge length, 3 mm in diameter, and 6 mm fillet radius at the edges and were measured according to ASTM E-8. Three samples were tested to confirm reproducibility, and their strength and elongation were averaged.

## 3. Results

### 3.1. Microstructure of Powder Mixtures and Composites

The morphologies of the as-milled powders with different Eu_2_O_3_ mass percentages are shown in Figure 2. The ball milling mixtures show large-size matrix and Eu_2_O_3_ particles with a small size stick on the surface of matrix particles when the contents of Eu_2_O_3_ are 15 wt.% and 20 wt.%, however, they almost disappear when the contents of Eu_2_O_3_ are 5 wt.% and 10 wt.%. The XRD patterns of the as-milled powders are shown in Figure 3. In Figure 3, no Eu_2_O_3_ diffraction peaks were detected after milling for 2 h when the contents of Eu_2_O_3_ were 5 wt.% and 10 wt.%, and there was only a diffraction hill near 30° indicating that the Eu_2_O_3_ powders conducted amorphous transformations. As the mass percentage of Eu_2_O_3_ increased to 15 wt.% and 20 wt.%, the characteristic Eu_2_O_3_ peaks appeared, and its intensity gradually increased.

During high-energy ball milling, Ni and Cr powder particles become finer since they are repeatedly flattened, cold-welded, and fractured [25,26]. Eu_2_O_3_ is a brittle powder, and the long-term milling process can provide the driving force for the Eu_2_O_3_ particle dissolution of Eu and O atoms into Ni/Cr crystal structure to form a supersaturated solid solution [27,28]. From Figure 2 and Figure 3, when the contents of Eu_2_O_3_ are 5 wt.% and 10 wt.%, they almost disappear on the surface of matrix, hence, Eu_2_O_3_ was fined, nano-crystallized, amorphized, and then mechanically dissolved into the matrix during the 2 h mechanical alloying process. Similar experiment phenomena were also observed in Pasebani’s report [29]. After 2 h of ball milling, the Y_2_O_3_ partial dissolution into the Ni-20Cr matrix and increasing matrix lattice parameter were observed in Pasebani’s research. Because of the limited solid solution, Eu_2_O_3_ partially dissolved into the matrix and partially stuck on the matrix surface when the contents of Eu_2_O_3_ were 15 wt.% and 20 wt.%.

The as-milled powders solidify after hot pressing. The experimental density, theoretical density, and relative density of the samples are shown in Table 1. The bulk relative density is 99.29%, 99.52%, 100%, and 100% of the theoretical value of HP5, HP10, HP15, and HP20, respectively. There are no pores observed, which indicates that hot pressing is an effective way to obtain high-density bulk material. Typical microstructure images of the Ni-20Cr-Eu_2_O_3_ composites with different Eu_2_O_3_ mass percentages are shown in Figure 4a–d, and higher magnification images are shown in Figure 4e–h. The Eu_2_O_3_ particles mainly existed in two different forms: nano-crystalline particles and agglomerated Eu_2_O_3_ with larger grain sizes. This is in consonance with the result of SEM observation as-milled powders had a broad size distribution of dispersoids containing both very refined and enlarged particles. It is observed that the size and number of the Eu_2_O_3_ clusters increase as the Eu_2_O_3_ concentration increases from 5 wt.% to 20 wt.%.

Table 2 summarizes the results of an EDS analysis of several representative areas of the HP15 to determine its chemical composition. The gray area (Area B, D) is the matrix composed of Ni and Cr, the black area (Area E, F) is the Cr enriched area, and the white area (Area A, C) is enriched with the elements Eu and O, as shown in Figure 4 and Table 2. The XRD patterns of the Ni-20Cr-Eu_2_O_3_ composites are shown in Figure 5. Only Eu_2_O_3_ and CrNi_3_ phase diffraction peaks in the XRD spectra following sintering. The lack of diffraction peaks relating to the alloy compounds between Eu_2_O_3_ and Ni or Cr indicates that there is no chemical reaction between Eu_2_O_3_ and Ni or Cr at the sintering temperature of 1250 °C.

In Figure 6, the rotation angles (RAs) of grain boundaries of Ni-20Cr-Eu_2_O_3_ composites are presented in green (2–5°), red (5–15°), and blue (15–180°) lines in image quality mappings detected by EBSD, and three kinds of RA percentages are shown in the inserted images. Here, low-angle grain boundaries (LAGBs) are defined as a misorientation of 2–15°, while large-angle grain boundaries (HAGBs, 15–180°) are defined as a misorientation of >15°. Gray and black areas indicate the phases of the matrix and Eu_2_O_3_, respectively. The fraction of HAGBs is the most common type of grain boundary in Ni-20Cr-Eu_2_O_3_ composites. It is believed that increasing the proportion of HAGBs is conducive to enhancing the strength of most alloys. When a crack propagates to the front of HAGBs, more energy must be overcome and consumed than when it propagates to the back of LAGBs, obstructing crack propagation. Furthermore, the relative frequency of the grain misorientation angle at 60° is high, which is ascribed to the generation of numerous twin boundaries (TBs) during the HP process. Annealing twins have been frequently observed during grain growth of nickel because of their low stacking fault energy, which was associated with thermal and sintering processes [30,31].

The grain size distributions of the composites are shown in Figure 7. Figure 7 shows that the average grain size increases with increasing Eu_2_O_3_ content, with HP5 having the smallest average grain size of about 2.2749 μm, the average grain sizes of HP10, HP15, and HP20 are 2.3473 μm, 2.4362 μm, and 2.5053 μm, respectively. Dispersed oxides are effective in retarding recovery and recrystallization during the sintering process by impeding dislocation glide and grain growth at elevated temperatures, a process known as Zener pinning [32]. Eu_2_O_3_ particles, as a sub-grain boundary nucleation source, can effectively prevent grain growth of the Ni-20Cr during solidification via the pinning effect, which can play a role in grain refinement in Ni-20Cr-Eu_2_O_3_ composites [33,34,35]. Higher oxide particle concentrations cause a slight increase in the grain size of the alloy in this study because an increase in agglomerated oxide particles reduces the number density of dispersed oxide particles.

### 3.2. TEM Analysis of Interfacial Structures

Figure 8 shows bright-field TEM images of the HP10 and HP20. It can be found that the matrix is an equiaxed structure, and the Eu_2_O_3_ particles are randomly distributed nanoparticles precipitated in grain interiors and on grain boundaries. All of the Ni-20Cr-Eu_2_O_3_ composites contained a high proportion of twin boundaries. EDS mapping is used to examine the chemical compositions of the composites. The EDS maps shown in Figure 8c indicate that the matrix composes of Ni and Cr, and the particles are enriched with the elements Eu and O and are depleted of Ni and Cr, along with a small Cr-rich area that is consistent with the SEM results. It is likely that during hot-pressing at elevated temperatures, Cr started precipitating out of the Ni-Cr solid solution in the form of Cr-based oxides, which is consistent with previously reported results [36].

Figure 9 depicts the TEM images, selected area electron diffraction (SAED) patterns, and a high-resolution transmission electron microscopy (HRTEM) image of the interface between the Ni matrix and the Eu_2_O_3_ particles in HP10. The full image of the matrix, twin boundary, and Eu_2_O_3_ particles is shown in Figure 9a. The SAED patterns of the matrix (A1 area) and twin boundary are shown in Figure 9b,c (A2 area). The interface between the Ni matrix and the Eu_2_O_3_ particles is visible in Figure 9d. The high-resolution image in Figure 9e is of the A3 area in Figure 9a, where the atom plane (200) of the matrix and ($\bar{5}$11) of Eu_2_O_3_ particles were detected. Twins and Eu_2_O_3_ interface structures were observed in HP20, as shown in Figure 10a, and their corresponding SAED patterns are shown in Figure 10b,c. HRTEM images of the typical Eu_2_O_3_ particle and matrix interface, as shown in Figure 10d,e, were captured in area B2. The interplanar distance of the Eu_2_O_3_ grains was 0.2937 nm, which corresponds to the (401) plane of the Eu_2_O_3_, and the interplanar distance of the matrix was 0.2143 nm, which corresponds to the (111) plane of the matrix, as shown in Figure 10e.

According to Figure 9 and Figure 10, all of the grains clearly showed lattice fringes on the boundary between the Eu_2_O_3_ particle and matrix, and there were no other impurities or amorphous phases, indicating that the composites had strong interface compatibility. Because of their similar lattice parameters, two adjacent crystals shared some of the atoms at the Eu_2_O_3_–matrix interface. This semi-coherent interface exhibited strong interfacial bonding between the Eu_2_O_3_ particle and matrix, with low interface energy. These interface structures also contributed to the composites’ high ultimate tensile strength [37,38].

### 3.3. Mechanical Properties

The Vickers hardness, ultimate tensile strength (UTS), yield strength (YS), and elongation values of the composites at room temperature are given in Table 3. The Vickers hardness of the composites increases as the mass percentage of Eu_2_O_3_ increases. This is due to the addition of the Eu_2_O_3_ particles to the Ni-20Cr matrix, which increases the resistance to local plastic deformation. By limiting the slip of dislocation, the reinforcing particles confine the matrix deformation to a restricted area, increasing residual stress and hardness of the composites [39,40,41].

The nominal tensile strain-stress curves of the Ni-20Cr-Eu_2_O_3_ composites are shown in Figure 11. As shown in Table 3 and Figure 11, HP5 shows a high ultimate tensile strength (UTS), yield strength (YS), and elongation, and the corresponding values are 741 MPa, 663 MPa, and 4%, respectively. The tensile strength was the lowest (556 MPa) when 20 wt.% Eu_2_O_3_ was added. With the increase in the Eu_2_O_3_ content, the tensile strength, yield strength, and elongation decreased gradually. The decrease in the value can be attributed to a higher concentration of Eu_2_O_3_ in the matrix, leading to more agglomeration of the Eu_2_O_3_ particles and weakening their mechanical bonding with the matrix [42].

Fracture morphologies of tensile specimens have been characterized and analyzed to investigate the underlying mechanisms of various mechanical properties. Figure 12 shows cross-sections of composites after tensile at room temperature, which show various fracture types. Dimples with typical plastic fracture features were observed in composites HP5 and HP10, indicating that they have higher plasticity than composites HP15 and HP20. At the bottom of dimples, nano-sized particles could be found, acting as the source of cracking during the tensile test. With the increase in the Eu_2_O_3_ content, a brittle surface with intergranular rupture is observed in composites HP15 and HP20.

### 3.4. Strengthening Mechanism in Ni-20Cr-Eu_2_O_3_ Composites

The mechanical properties of composites are strongly influenced by microstructure parameters such as particle distribution, solid solution, dislocation density, and grain size. The generation of dislocations as a result of the different coefficient of thermal expansion (CTE) between the Ni matrix and the reinforcements is generally ascribed to the thermal mismatch mechanism in composites; Grain refining caused by the pinning effect of Eu_2_O_3_ particles; Solid-solution strengthening is attributed to the dilute solution alloys. Hence, $\sigma_{Y}$ it can be estimated by the following equation:(1)$$\sigma_{Y}=\sigma_{m}+\Delta \sigma_{CTE}+\Delta \sigma_{GB}+\Delta \sigma_{ss}$$
where $\sigma_{Y}$ and $\sigma_{m}$ are the estimated yield strength of Ni-20Cr-Eu_2_O_3_ composites and matrix, respectively [43]. $\Delta \sigma_{CTE}$ is the thermal mismatch contribution, $\Delta \sigma_{GB}$ is the grain boundary strengthening contribution, and $\Delta \sigma_{SS}$ is the solid solution strengthening contribution.

#### 3.4.1. Thermal Mismatch Strengthening

Due to the large difference in the CTE between the particles ($10\times 10^{-6}K^{-1}$ for Eu_2_O_3_) and matrix ($13.4\times 10^{-6}K^{-1}$ for Ni), new dislocations may generate on the interface during cooling down, thus increasing the dislocation density [44]. This thermal mismatch mechanism in Ni-based composites is related to the dislocation density generated during the process, and the contribution can be estimated using the following equation:(2)$$\Delta \sigma_{CTE}=\alpha MGb\rho^{1/2}$$
where *G* (shear modulus), *b* (Burgers vector), and *M* (Taylor factor) were taken as 82 GPa, 0.25 nm, and 3, and the dislocation strengthening coefficient α was utilized as 0.25 as proposed by Sevillano for fcc crystals with high dislocation density [45,46]. Equation (3) shows the calculation of the enhanced dislocation density (ρ) in the Ni-based composites:(3)$$\rho =\frac{12V_{p}\Delta \alpha \Delta T}{(1-V_{p})bd_{p}}$$
where Δα represents the difference in CTE between Ni-20Cr matrix and particles, ΔT is the difference in temperature between the sintering process (1250 °C) and the ambient temperature during the tensile test (25 °C), $V_{p}$ and $d_{p}$ are the volume fraction and approximate diameter of particles, where $d_{p}$ obtained by EBSD. The calculated thermal mismatch strengthening values are shown in Table 4.

#### 3.4.2. Grain Boundaries Strengthening

The grain boundary strengthening contribution is estimated using the classic Hall-Petch relationship. EBSD was used to determine the average grain size for the various Ni-based Eu_2_O_3_ composites, which is presented in the results section. By substituting the obtained average grain sizes (d, nm) in the equations as suggested by Bui et al. [47].
(4)$$\Delta \sigma_{GB}=\left(5538{\mathrm{MPanm}}^{1/2}\right)d^{-1/2}$$

#### 3.4.3. Solution Strengthening

The solid solution effects brought on by member alloying additions in Ni-based alloys can be calculated using the relation below:(5)$$\Delta \sigma_{SS}={⟮\sum_{i}k_{i}^{1/n}c_{i}⟯}^{n}$$
where $\Delta \sigma_{SS}$ is the solid solution contribution, $k_{i}$ is the strengthening constant for solute element *i* in Ni, $c_{i}$ is the concentration of solute *i*, and *n* is taken as 0.5 here. For Ni-Cr alloy, the value of k is 337 MPa at fraction-1/2 and assuming that the particles did not get dissolved in the matrix [48,49].

Table 5 summarizes the contribution of various strengthening mechanisms in Ni-20Cr-Eu_2_O_3_ composites and compares estimated yield strength values to experimentally determined yield strength values. Thermal mismatch strengthening produces relatively small theoretical strength gains when compared with grain boundary strengthening and solution strengthening. The tendencies of values, however, show a constant increase with the increase of reinforcement content. However, the contribution did not vary significantly with increasing Eu_2_O_3_ concentration because these composites have very little variation in grain size, as shown in Figure 7. The results show that the experimental strengths of composites HP5 and HP10 are well matched with the theoretical values indicating that the method is highly effective for predicting the mechanical properties of Ni-20Cr-Eu_2_O_3_ composites. It was also discovered that when the Eu_2_O_3_ content of the composites exceeded 15%, the yield strength of the composites decreased dramatically, owing primarily to the formation of Eu_2_O_3_ clusters, which affect effective stress transfer during tensile deformation. This basically means that the volume fraction of Eu_2_O_3_ particles in these composites has a big impact on the interfacial interactions and strengthening efficiency.

## 4. Conclusions

Ball milling and hot pressing are used to prepare Ni-20Cr-Eu_2_O_3_ composites in this paper. The microstructure evolution of Eu_2_O_3_ during ball milling is investigated, as well as the effects of Eu_2_O_3_ on the density, phase composition, and grain size of composites. Mechanical properties are evaluated and quantified using various strengthening mechanisms. The following are the main conclusions:

(1)During ball milling, Eu_2_O_3_ powders were continuously refined. When the Eu_2_O_3_ content was less than 10 wt.%, almost all of the Eu_2_O_3_ particles dissolved in the matrix; when the Eu_2_O_3_ content was greater than 10 wt.%, some Eu_2_O_3_ particles dissolved in the matrix, and some Eu_2_O_3_ particles agglomerated on the matrix surface as nanocrystals. The higher the content of Eu_2_O_3_, the higher the proportion of the agglomerated blocks.(2)High-density samples are obtained at 1250°C, and the composites mainly contain three different phases: the matrix phase, CrNi_3_, a black Cr-rich region, and Eu_2_O_3_ in the form of dispersed particles and agglomerates. The average grain size increases with increasing Eu_2_O_3_ content, with HP5 having the smallest average grain size of about 2.2749 μm, the average grain sizes of HP10, HP15, and HP20 are 2.3473 μm, 2.4362 μm, and 2.5053 μm, respectively.(3)The Eu_2_O_3_ particles are well bonded to the Ni-20Cr matrix and a semi-coherent interface forms between the Ni-20Cr matrix and the Eu_2_O_3_ particles.(4)The values of the ultimate tensile strength for HP5, HP10, HP15 and HP20 are 741 MPa, 692 MPa, 600 MPa and 556 MPa, respectively. The yield strength and elongation decreased gradually as the Eu_2_O_3_ content increased. When 20 wt.% Eu_2_O_3_ was added, and the tensile strength was the lowest (556 MPa).

## Figures and Tables

**Figure 1 materials-16-01473-f001:**
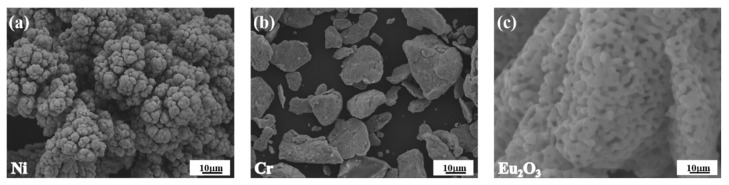
SEM images of raw materials:(**a**) Ni; (**b**) Cr; (**c**) Eu_2_O_3_.

**Figure 2 materials-16-01473-f002:**
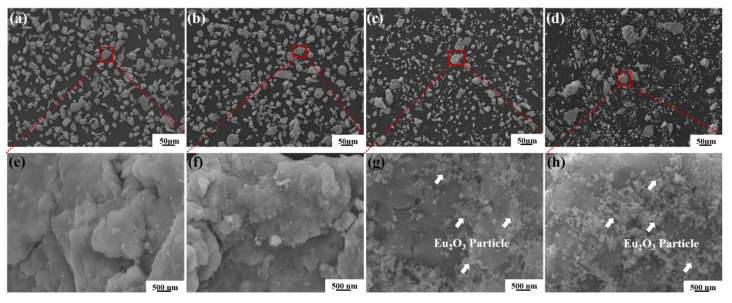
SEM images of the as-milled powders with different Eu_2_O_3_ mass percentages: (**a**,**e**) HP5; (**b**,**f**) HP10; (**c**,**g**) HP15; (**d**,**h**) HP20.

**Figure 3 materials-16-01473-f003:**
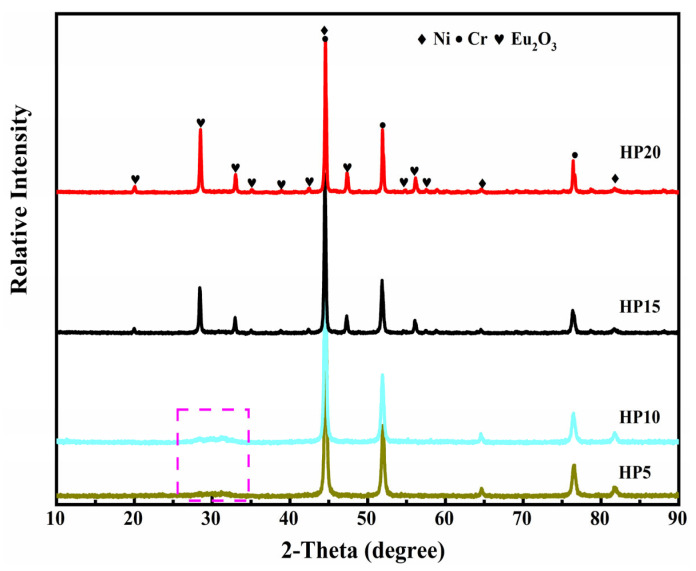
XRD patterns of as-milled powders.

**Figure 4 materials-16-01473-f004:**
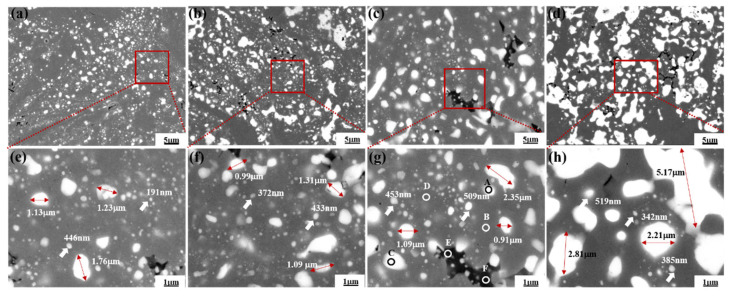
SEM images of the composites with different Eu_2_O_3_ mass percentages: (**a**,**e**) HP5; (**b**,**f**) HP10; (**c,g**) HP15; (**d**,**h**) HP20.

**Figure 5 materials-16-01473-f005:**
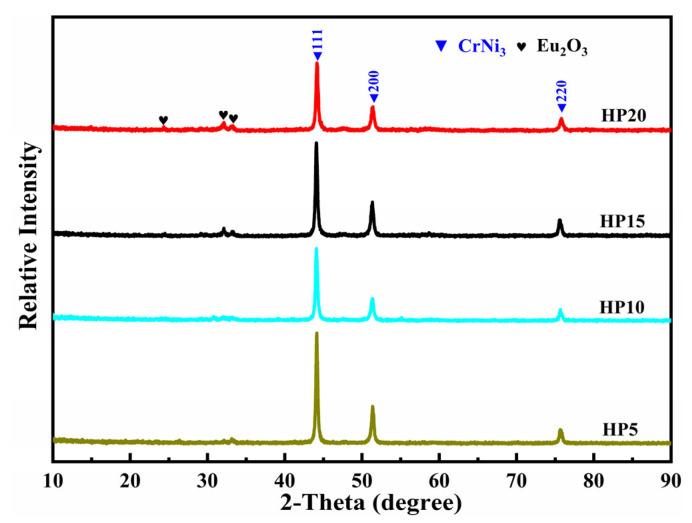
XRD patterns of Ni-20Cr-Eu_2_O_3_ composites.

**Figure 6 materials-16-01473-f006:**
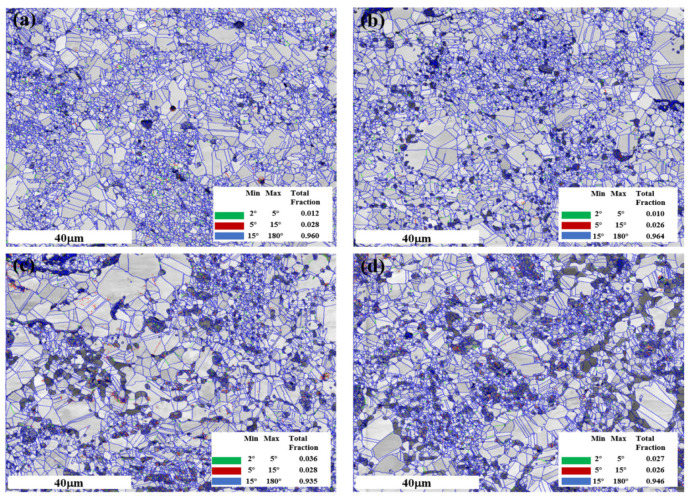
Grain information of the composites associated with (**a**) HP5; (**b**) HP10; (**c**) HP15; (**d**) HP20.

**Figure 7 materials-16-01473-f007:**
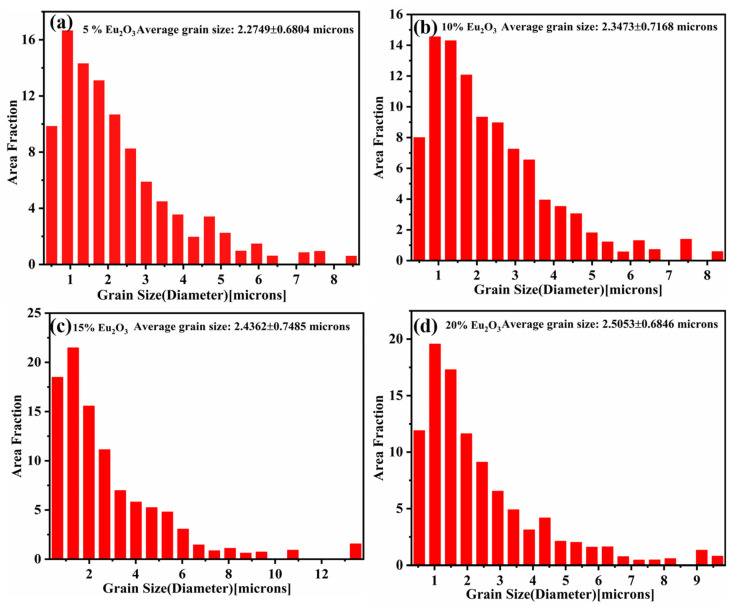
Grain size distribution of the composites: (**a**) HP5; (**b**) HP10; (**c**) HP15; (**d**) HP20.

**Figure 8 materials-16-01473-f008:**
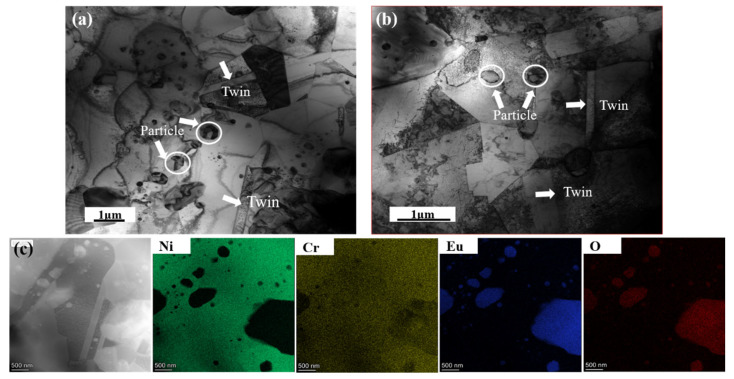
TEM images with the corresponding EDS mappings of the composites: (**a**) a bright TEM image of HP10; (**b**) a bright TEM image of HP20; (**c**) the EDS mapping of HP10.

**Figure 9 materials-16-01473-f009:**
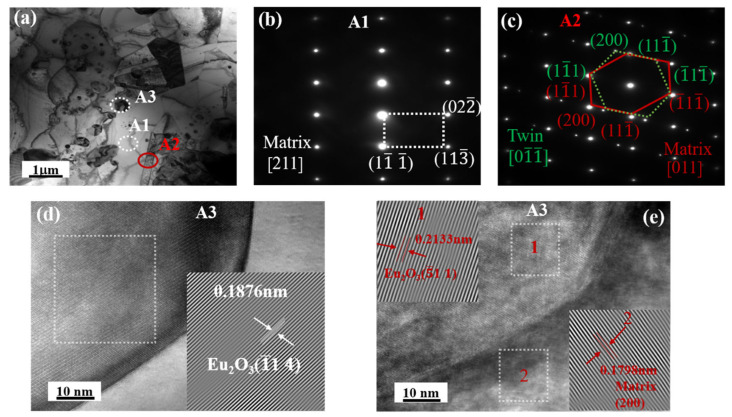
TEM bright field images of the HP10 composite: (**a**) A bright TEM image; (**b**) the corresponding SAED patterns of matrix; (**c**) the corresponding SAED patterns of twin; (**d**) obtained from the regions marked in A3 in (**a**); (**e**) obtained from the regions marked in 1 and 2 in (**d**).

**Figure 10 materials-16-01473-f010:**
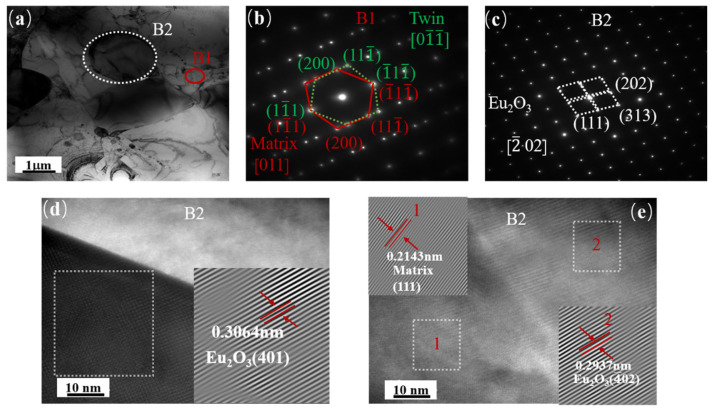
TEM bright field images of the HP20 composite: (**a**) A bright TEM image; (**b**) the corresponding SAED patterns of twin; (**c**) the corresponding SAED patterns of Eu_2_O_3_; (**d**) obtained from the regions marked in B2 in (**a**); (**e**) obtained from the regions marked in 1 and 2 in (**d**).

**Figure 11 materials-16-01473-f011:**
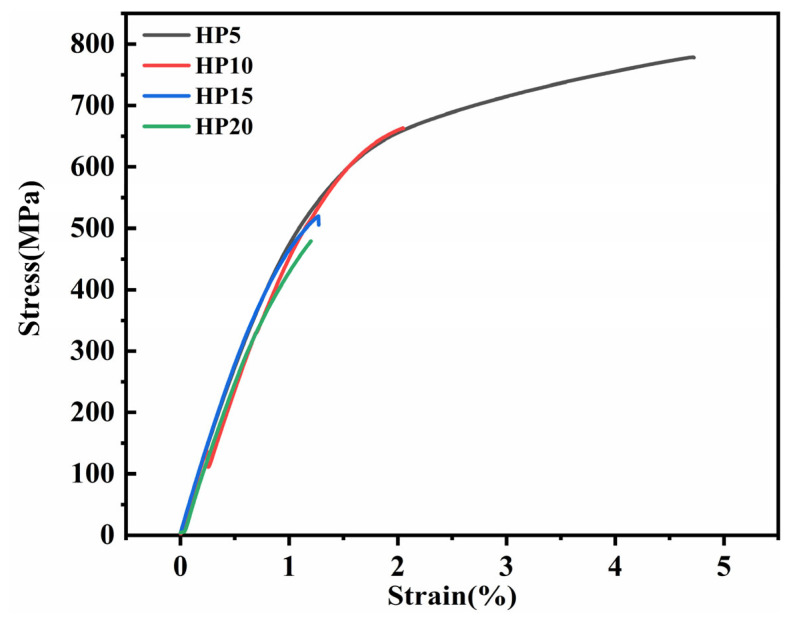
Engineering stress-strain curves of Ni-20Cr-Eu_2_O_3_ composites.

**Figure 12 materials-16-01473-f012:**
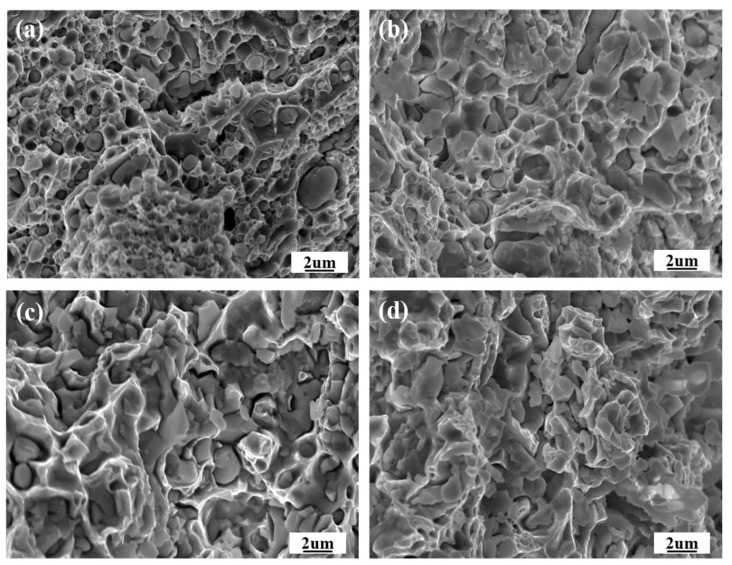
FESEM images of the fracture morphology after tensile tests: (**a**) HP5; (**b**) HP10; (**c**) HP15; (**d**) HP20.

**Table 1 materials-16-01473-t001:** The composition (wt.%), experimental density, theoretical density, and relative density of specimens.

Sample’s Code	Cr	Eu_2_O_3_	Ni	Experimental Density	Theoretical Density	Relative Density (%)
HP5	20	5	Bal.	8.35 ± 0.36	8.41	99.29
HP10	20	10	Bal.	8.28 ± 0.24	8.32	99.52
HP15	20	15	Bal.	8.26 ± 0.15	8.26	100
HP20	20	20	Bal.	8.26 ± 0.12	8.26	100

**Table 2 materials-16-01473-t002:** Chemical composition (wt.%) at each point in HP15 composite.

Elements	Gray Area		White Area		Black Area	
	B	D	A	C	E	F
Ni	76.17	72.08	3.17	2.68	28.35	9.70
Cr	21.73	19.76	7.22	7.18	67.71	88.03
Eu	2.10	8.16	72.05	72.49	3.94	2.28
O	/	/	17.55	17.66	/	/

**Table 3 materials-16-01473-t003:** Mechanical properties of the Ni-20Cr-Eu_2_O_3_ composites.

Composites	Ultimate Tensile Strength (MPa)	Yield Strength (MPa)	Elongation (%)	Hardness (HV)
HP5	741	663	4	281
HP10	692	659	2	298
HP15	600	574	1	308
HP20	556	/	/	312

**Table 4 materials-16-01473-t004:** Parameters and strengthening contribution from thermal mismatch strengthening mechanisms of the Ni-20Cr-Eu_2_O_3_ composites with different mass ratios in Eu_2_O_3_ (MPa).

Composition	Content of Reinforcements/vol.%	$\rho \times 10^{6}$	$\Delta \sigma_{CTE}/\mathrm{MPa}$
HP5	5.67	3.7	58
HP10	11.24	5.21	80
HP15	16.71	5.31	81

**Table 5 materials-16-01473-t005:** Contribution of different strengthening mechanisms in the Ni-20Cr-Eu_2_O_3_ composites (MPa).

Composite	$\sigma_{m}$	$\Delta \sigma_{CTE}$	$\Delta \sigma_{GB}$	$\Delta \sigma_{SS}$	$\sigma_{Y}$	$\sigma_{Exp}$	Difference/%
HP5	359	58	116	75	608	662	8.15
HP10		80	114	107	660	679	2.80
HP15		81	104	131	675	574	17.60

## Data Availability

The data presented in this study are available upon request from the corresponding author.

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
