# Peer review of "Microstructure and Mechanical Properties of Ni-20Cr-Eu2O3 Composites Prepared by Vacuum Hot Pressing"

_materials, 2023, doi:10.3390/ma16041473_

Round 1

Author Response

Response to Reviewer 1 Comments

Point 1: (Page 1, Line 16-17) “The relative densities, grain size, and microhardness were positively correlated with the content of Eu2O3, opposite for the twin volume fraction”. This statement is not fully supported by the results presented in the subsequent sections.

Response 1: The statement has been revised to “The relative densities, grain size, and microhardness increased with Eu2O3 content increased.”

Point 2: (Page 1, Line 23-24) “And the differences between them are less than 8.5%,indicating that this method is highly effective for predicting the mechanical properties of Ni-20Cr-Eu2O3 composites.” This statement needs further qualification as the methodology shows only less than 8.5% difference for two cases. The authors need to provide more evidence to make this claim for Ni-20Cr-Eu2O3 composites in general.

Response 2: The abstract section has been rewritten.

Point 3: (Page 1, Line 47-49) “Eu has a good absorption cross-section, moreover, the Euisotope produced after irradiation has excellent neutron absorption capacity for a long lifetime, and it has been used in the Russian BOR-60 and BN-600 reactors. Eu will not be used in a reactor in the metallic state, since the metal is extremely reactive toward water and oxygen.” Please add the reference for this.

Response 3: The reference has been added.

Point 4: (Page 2, Line 59-60) “Meanwhile, with developments in nuclear reactors, such as VHTR and GFR reactors in Generation IV, the operation temperatures are much higher than the reactors of the past generations [13].” The Introduction needs to be improved by including why there is a need to move to a new generation of nuclear reactors.

Response 4: The “Meanwhile, with developments in nuclear reactors, such as VHTR and GFR reactors in Generation IV, the operation temperatures are much higher than the reactors of the past generations [13]” section has been rewritten.

Point 5: (Page 2, Line 65-66) “Nosov et al. prepared Ni-based rare earth (RE) oxides (RE=Sm, Eu, Gd, Dy, Er) composites with high physical efficiency and good thermophysical properties [20].” Please add the key findings of this reference and how they are relevant to the current study.

Response 5: The “Nosov et al. prepared Ni-based rare earth (RE) oxides (RE=Sm, Eu, Gd, Dy, Er) composites with high physical efficiency and good thermophysical properties [20].” section has been rewritten.

Point 6: There are a few other studies which consider addition of oxides specifically to the NiCr matrix
https://link.springer.com/article/10.1007/BF02153458
https://www.jstage.jst.go.jp/article/matertrans1960/24/12/24_12_839/_article/-char/ja/

There are also extensive studies on Ni-based alloys in the Oxide Dispersed Strengthening (ODS) field.
https://www.sciencedirect.com/science/article/pii/S0010938X21008465
https://www.sciencedirect.com/science/article/pii/S0921509318313315

Response 6: The reference has been added.

Point 7 There are far too many figures in the Main Text. I would recommend that some of the figure be moved to the Supplementary Information file for better readability. 

Response7: It is true that there are more diagrams in the text, but most of them are intended to illustrate the relevant issues, and if added to the supplementary document may affect its overall.

Point 8: The authors switch between using (HP5, HP10, HP15 and HP20) and (5, 10, 15, and20 wt.%) Eu2O3 notation for the composites. I would recommend adopting either one of the styles throughout the manuscript.

Response 8: I has been adopting one of the styles (HP5, HP10, HP15, HP20) throughout the manuscript.

Point 9: (Page 2, Line 71,72)“Therefore, Eu2O3 with different mass fractions were used to fabricate Ni-20Cr-Eu2O3composites via ball milling and vacuum hot-pressing method in this study.” The authors need to discuss why this methodology was chosen over other methods such as conventional sintering or spark plasma sintering.

Response 9: The discuss has been added.

Point 10: (Page2, Line 74,75) The phase composition, grain size distribution, crystal structure, and misorientation relationship of Ni-20Cr-Eu2O3 composites were systematically analyzed.” This statement needs to be rephrased as the phase composition using EDS is only shown for the HP15 composite in Fig.5. Please add the corresponding composition analysis for all the composites to the SI to support this statement.

Response 10:

Fig. 1 Chemical composition (wt.%) at each point in composite: (a) HP5; (b) HP10;(c) HP15;(d) HP20

Chemical composition (wt.%) at each point in HP5 composite

Elements

Gray area

White area

Black area

B1

A1

C1

Ni

77.55

12.79

29.53

Cr

20.13

4.98

44.38

Eu

2.32

66.95

26.09

O

/

15.28

/

Chemical composition (wt.%) at each point in HP10 composite

Elements

Gray area

White area

Black area

B2

A2

C2

D2

E2

Ni

74.36

2.78

10.59

9.54

8.75

Cr

21.13

8.63

5.27

90.46

91.25

Eu

4.51

71.11

68.91

/

/

O

/

17.48

15.22

/

/

Chemical composition (wt.%) at each point in HP15 composite

Elements

Gray area

White area

Black area

B3

D3

A3

C3

E3

F3

Ni

76.17

72.08

3.17

2.68

28.35

9.70

Cr

21.73

19.76

7.22

7.18

67.71

88.03

Eu

2.10

8.16

72.05

72.49

3.94

2.28

O

/

/

17.55

17.66

/

/

Chemical composition (wt.%) at each point in HP20 composite

Elements

Gray area

White area

Black area

B3

A3

C4

D4

Ni

75.32

2.05

24.82

35.44

Cr

23.99

12.39

75.18

64.56

Eu

0.69

67.95

/

/

O

/

17.62

/

/

The results of their EDS analysis show that the four composites have the same physical phase composition, so one of them is chosen to illustrate the problem.

Point 11: (Page 3, Line 102-103) “The physical density of the hot-pressed samples was measured using Archimedes’ principle with at least six measurements for each sample.” Please add the corresponding error bars to Table 1.

Response 11: The standard deviation has been added.

Point 12: Table 1: Please mention how the theoretical density was calculated in the Table 1 caption. Response 12:

The theoretical density of the material is:

Samples

theoretical density

Eu2O3:7.42g/cm3;Ni:8.90g/cm3;Cr:7.20g/cm3

HP5

8.41

HP10

8.32

HP15

8.24

HP20

8.16

In section 2.2, this sentence ‘’The densities of Ni, Cr and Eu2O3 used for theoretical densities calculations were 8.90 g/cm3, 7.20 g/cm3, 7.42 g/cm3, respectively.’’ was added

Point 13: (Page 3, Line 105-108) “The mechanical properties, including yield strength, tensile strength and elongation was measured by a universal material testing machine (Shimadzu AG-250KNIS). The samples were cut into dog bone-shaped specimens of 35 mm in gauge length, 3 mm in diameter, and 6 mm fillet radius at edges.” Please indicate how many samples were tested in each case.

Response 13: The number of the sample has been added.

Point 14: (Page 3, Line 126-128) “The Eu2O3 is brittle powder and the long-term milling process can provide the driving force for the Eu2O3 particles dissolution of Eu and O atoms into Ni/Cr crystal structure to form a supersaturated solid solution [23, 24]”.

  1. While this could be true, it does not follow from Ref [23] dealing with Mo-based Tb2O3-Dy2O3 composites and Ref [24] which involves the Mo-Tm2O3. The authors need to address the solubility of Eu2O3 specifically into the Ni 20Cr matrix as the solubilities on Mo in Tm2O3 and Tb2O3-Dy2O3 in Mo could be different from that of Eu into the Ni-20Cr matrix.
    b. Figure 3: Although, the XRD patterns of as-milled powders of the 5 and 10 wt% Eu2O3 do not show the peaks corresponding to Eu2O3, this could also bedue to the sensitivity limit of XRD.  

Response 14: a: The current research on rare earth oxides in composite or ODS alloy is mainly on Y2O3, La2O3, Tb2O3 and Dy2O3, but less on Eu2O3. The citation Mo-based Tb2O3-Dy2O3 composites and Mo-Tm2O3 is to show that Eu2O3, as a rare earth oxide particle, also undergoes decomposition reactions like Tb2O3 and Dy2O3. Meanwhile, in ODS alloys, there is a similar occurrence of Y2O3.

b:XRD is generally tested at contents above 5%, while, as seen in Figure 2, there are almost no oxide particles on the surface at 5 wt% and 10 wt%, which also indicates the possibility of decomposition reactions.

Point 15: (Page 4, Line 155) “Several representative areas of the HP15 are chosen to analyze the chemical composition by EDS, and the values are summarized in Table 2.” Please add error bars for the chemical compositions reported in Table2.

Response 15: Since the EDS test itself is a qualitative result, error bars are not generally provided in the literature for it.

Point 16: (Page 4, Line 157-159) “enriched area, and the white area (Area A, C) is enriched with the elements of Eu and O. The XRD patterns of the Ni-20Cr-Eu2O3 composites was shown in Fig. 3”. Please correct this to Fig.5.

Response 16: The fig.3 has been modified into fig. 5

Point 17:. (Page 6, Line 193-195) “In this study, higher oxide particle concentrations result in a slight increase in the grain size of the alloy due to an increase in agglomerated oxide particles decreases the number density of dispersed oxide particles.” In Fig.7, it is shown that the grain size increases even for the 5wt% Eu2O3 case compared to the 10wt% Eu2O3 case from 2.27 microns to 2.34 microns. Does this mean that to achieve grain refinement even lower wt% of Eu2O3 must be used? The authors need to discuss this further.

Response 17: Not that to achieve grain refinement even lower wt% of Eu2O3 must be used, it is the agglomeration of oxide particles that occurs as the Eu2O3 content increases, which leads to a decrease in the number density of its fine-grained reinforced nanoparticles, which in turn affects the effect of grain refinement, and that is why it leads to a slight increase in grain size as the oxide particles increase.

Point 18: Figure 7: Please add error bars to the calculated average values.

Response 18: The error bars has been added.

Point 19: (Page 7, Line 202)”Fig. 8 shows bright-field TEM images of the Ni-20Cr-Eu2O3 composites containing different contents of Eu2O3 particles.” Please add which composites are being shown in the figure.

Response 19: The statement has been revised to “Fig. 8 shows bright-field TEM images of the HP10 and HP20”

Point 20: (Page 7, Line 209-211)” It is likely that during hot pressing at the elevated temperatures, Cr started precipitating out of the Ni-Cr solid solution in the form of Cr-based oxides, which was consistent with previously reported results.” Please addthe reference here.   

Response 20: The reference has been added.

Point 21: (Page 8, Line 228)” According to Fig. 9 and 10, all of the grain clearly showed lattice fringes on the boundary between the Eu2O3 particle and matrix, and there were no other impurities or amorphous phases, indicating that the composites had strong interface compatibility.” The statement is not supported by Fig.9 and 10 which have only TEM images reported for the HP10 and HP20 composites. The authors need to do a systematic analysis of all the composites to make this statement.

Response 21: The analysis of their XRD, SEM, and EDS results shows that the four composites have the same physical phase composition and composition, except for their Eu2O3 content, and two of them are selected to have representative results.

Point 22: Table 5: Please the reference for the estimation of ?? in the text.

Response 22: The reference has been added.

Point 23: (Page 12, Line 332-33)” All Ni-20Cr-Eu2O3 composites demonstrated a significant improvement in yield strength due to grain refinement strengthening due to the addition of Eu2O3 particles in the matrix.” This claim is seemingly the most important result, but it is not substantiated with all the required evidence. The authors need to compare the strengthening effect of Eu2O3 relative to the matrix of Ni-20Cr without any addition of Eu2O3 as a control experiment. The authors must compare the grain size of the Ni-20Cr without any additions to the current set of composites to claim that the strengthening is due to grain refinement.

Response 23: In this study, the main focus was to investigate the effect of four different Eu2O3 contents on the microstructure and mechanical properties of the composites. It has been extensively reported in the literature that the increase of oxide particles has a grain refining effect on the alloy. As for the effect on its mechanical properties, the ?? in the text means the Ni-20Cr without any additions has shown that the addition of Eu2O3 has an effect on the addition of its mechanical properties.

Point 24: (Page 12, Line 356) “and some Eu2O3 particles agglomerated on the matrix surface as nanocrystals” It is not clear that the agglomerates are nanocrystals. From the SEM images in Fig.4, the agglomerates appear to be micron sized. The authors need to provide higher resolution images of the agglomerates to conclude that there are nanocrystalline agglomerates.

Response 24: What is illustrated here is the particle size of Eu2O3 in the ball-milled powder, and it can be found from Figure 2 that the particle size of Eu2O3 on its surface is much lower than 500 nm at a scale of 500 nm, thus indicating the formation of nanoparticles

Point 25: (Page 13, Line 365) “The Eu2O3 particles are well bonded to the Ni-20Cr matrix and a semi-coherent interface form between the Ni-20Cr matrix and the Eu2O3 particles.” This statement is not fully supported by the TEM results presented in the manuscript where only HP20 and HP10 are shown. The authors need to systematically analyze all the composites before making this statement.

Response 25: The analysis of their XRD, SEM, and EDS results shows that the four composites have the same physical phase composition and composition, except for their Eu2O3 content, and two of them are selected to have representative results.

Point 26: The conclusions section in lacking in identifying a design strategy from the current study which would help future studies in the field. The authors also need to include future directions on how this study could be improved.

Response 26: Thank you for your suggestion, have made changes in the conclusion section

Reviewer 2 Report

This article deals with the effect of the amount of Eu2O3 phase on the mechanical properties and microstructure of composites produced by hot pressing method. Before the article is proposed for publication, it needs a significant improvement, which I have given below my comments.

First of all, the English language of the article should be revised. Many grammatical and structural errors need to be fixed.

The title of the article is incomplete. It should be as follows: “Microstructure and mechanical properties of Ni-20Cr-Eu2O3 composites fabricated/manufactured/produced by vacuum hot pressing”

A space should be left between words and reference numbers, e.g., “material [2]”.

Lines 51-53; “Therefore, ceramics or metal matrix composites (MMCs) such as europium oxide (Eu2O3) pellets, Eu2O3-HfO2/ZrO2, and metal-based composites are selected as control rod materials[9-12].” please remove "metal-based composites". it is the repetition of one word in the sentence.

Line 80; Please mention the name of the manufacturer of raw powders.

Line 84; “jar” is a more common word than “can”.

Line 85; Please put a space between numbers and physical units, e.g., “400 r/min”. Please check this for all text.

Line 86; Please state the name of the manufacturer, city, and country from where the sintering equipment was sourced.

Please mention the material and dimensions of the die used as well as the dimensions of the produced samples.

Line 91; Please state the name of the manufacturer, city, and country from where the equipment was sourced. Please check it for all devices used in the study.

Lines 103-104; In this work, the samples are only produced by the hot pressing method, for which you have already provided the relevant explanation. Therefore, re-emphasis is considered redundant. In the whole text, just use “sample or samples” instead of “hot-pressed samples”.

lines 105-108; Please mention the standard code (e.g., ASTM_XXXX) you used to measure the mechanical properties of the samples.

Table 1; You have claimed to have measured the density of each sample 6 times using the Archimedes method. Why does your data have no standard deviation? Please add.

Lines 115-123; What is the reason for this long milling process that caused the ceramic powder to become amorphous? Does amorphous ceramic have an advantage over crystalline ceramic?

Line 142; What do you mean by this sentence “After hot pressing, the as-milled powders are solidified.”? The as-milled powders were not liquid that solidified! As I understood, you meant that the powders were consolidated. Am I right?

Line 159; The XRD patterns of samples after sintering are shown in fig. 5, not fig. 3. Please correct it.

Lines 159-162; Based on the XRD results, a gross mistake has occurred in the entire article. The XRD results show that nickel and chromium have reacted together during the sintering process and formed the new phase of CrNi3. In this case, the hot pressing is not a simple sintering process but is a reactive hot pressing. Moreover, the resulting composite is no longer Ni-20Cr-Eu2O3, but Ni3Cr-Eu2O3 composite, and no longer Metal-based composites. Note that the results presented in this article are for the mechanical properties as well as the microstructure of the samples produced after the sintering process and not the powders obtained after ball milling. Also, you should name the samples according to the chemical composition after sintering and not before. Realizing this, you should do a review from the title of the article to the conclusion.

Author Response

Response to Reviewer 2 Comments

Point 1: English language of the article should be revised. Many grammatical and structural errors need to be fixed.

Response 1: The English language of the article has been revised.

Point 2: Microstructure and mechanical properties of Ni-20Cr-Eu2O3 composites fabricated/manufactured/produced by vacuum hot pressing.

Response 2: The title has been revised ‘’Microstructure and mechanical properties of Ni-20Cr-Eu2O3 composites prepared by vacuum hot pressing’’.

Point 3: A space should be left between words and reference numbers, e.g., “material [2]”.

Response 3: The whole text has been put a space between words and reference numbers.

Point 4: Therefore, ceramics or metal matrix composites (MMCs) such as europium oxide (Eu2O3) pellets, Eu2O3-HfO2/ZrO2, and metal-based composites are selected as control rod materials [9-12].” please remove "metal-based composites".

Response 4: The metal-based composites has been removed.

Point 5: Please mention the name of the manufacturer of raw powders.

Response 5: The manufacturer of raw powders has been added.

Point 6: “jar” is a more common word than “can”.

Response 6: The ‘’can’’ has been revised ’’jar’’.

Point 7: Please put a space between numbers and physical units, e.g., “400 r/min”. Please check this for all text.

Response 7: The whole text has been put a space between numbers and physical units.

Point 8: Please state the name of the manufacturer, city, and country from where the sintering equipment was sourced.

Response 8: The name of the manufacturer, city, and country from where the sintering equipment was sourced was added.

Point 9: Please mention the material and dimensions of the die used as well as the dimensions of the produced samples.

Response 9: The material and dimensions of the die used as well as the dimensions of the produced samples was added.

Point 10: Please state the name of the manufacturer, city, and country from where the equipment was sourced. Please check it for all devices used in the study.

Response 10: The name of manufacturer, city and country from the equipment was sourced has been added.

Point 11: In this work, the samples are only produced by the hot pressing method, for which you have already provided the relevant explanation. Therefore, re-emphasis is considered redundant. In the whole text, just use “sample or samples” instead of “hot-pressed samples”.

Response 11: The whole text has been revised to sample or samples.

Point 12: Please mention the standard code (e.g., ASTM_XXXX) you used to measure the mechanical properties of the samples.

Response 12: The standard code has been added.

Point 13: Table 1; You have claimed to have measured the density of each sample 6 times using the Archimedes method. Why does your data have no standard deviation? Please add.

Response 13: The standard deviation has been added.

Point 14: Lines 115-123; What is the reason for this long milling process that caused the ceramic powder to become amorphous? Does amorphous ceramic have an advantage over crystalline ceramic?

Response 14: Thank you very much for your careful reading. Since Eu2O3 is friable, the ball milling process will lead to its refinement, which will further lead to its amorphization as the ball milling progresses. There are few studies related to Eu2O3, but common ones such as Y2O3 in ODS steels are reported more about its amorphization, and their related literature is as follows.

https://doi.org/10.1016/j.matchemphys.2012.08.056

https://doi.org/10.1016/j.msea.2015.01.066

https://doi.org/10.1016/j.apt.2015.08.017

Becoming amorphous means that the whole powder has been fully mixed during the ball milling process

Point 15: Line 142; What do you mean by this sentence “After hot pressing, the as-milled powders are solidified.”? The as-milled powders were not liquid that solidified! As I understood, you meant that the powders were consolidated. Am I right?

Response 15: Yes, it meant that the powders were consolidated.

Point 16: Line 159; The XRD patterns of samples after sintering are shown in fig. 5, not fig. 3. Please correct it.

Response 16: The fig.3 has been modified into fig. 5

Point 17: Lines 159-162; Based on the XRD results, a gross mistake has occurred in the entire article. The XRD results show that nickel and chromium have reacted together during the sintering process and formed the new phase of CrNi3. In this case, the hot pressing is not a simple sintering process but is a reactive hot pressing. Moreover, the resulting composite is no longer Ni-20Cr-Eu2O3, but Ni3Cr-Eu2O3 composite, and no longer Metal-based composites. Note that the results presented in this article are for the mechanical properties as well as the microstructure of the samples produced after the sintering process and not the powders obtained after ball milling. Also, you should name the samples according to the chemical composition after sintering and not before. Realizing this, you should do a review from the title of the article to the conclusion.

Response 17: Thank you very much for your careful reading and for your valuable comments. Regarding your question, first of all, the composite is mainly NiCr3 phase, but also some Cr phase, if only NiCr3 named the composite, it is not quite comprehensive. At the same time, in the naming of composite materials, some of the instrument alloy grades are named with oxide particles, as in the following related literature. In this study, the composite material is named by the original component because Ni-20Cr is not available.

Reviewer 3 Report

Reviewer Comment
Manuscript number: Materials- 2159792

Dear Editor,

The authors made a visible effort to investigate the effect of Eu2O3 addition on the mechanical, thermal and physical properties of Ni-20Cr alloy.

The submitted version of the manuscript has strengths and weaknesses. The most obvious weaknesses are:

1-The title should reflect the effect (influence) , (impact) of addition of Eu2O3 ceramic on the mechanical and microstructure of Ni-Cr alloy.

2-English language needs moderate editing. Especially, from Abstract and Introduction sections. Sentence rearrangement is also necessary for better understanding.

3-Abstract section should be rewritten, just for example, line 19-21.

4-The manufacturer (provider) names for Ni, Cr and Eu2O3 powders are not specified in section 2.1.

5-what does Bal. mean in Table 1?

6-EDX analysis described in line 154-157 should be added.

7-line 257: ... Eu2O3 ....Line 368

8-Equations in the manuscript should be written by the mathematical identifier in Word (if the authors applied the manuscript using Microsoft Word). The terms in section 3.4 should be rewritten through mathematical identifier too.

9-Based on the obtained outcomes of ultimate tensile strength, yield strength, elongation % and hardness, which composition is the optimum and will be preferred for the desired application?

Author Response

Response to Reviewer 3 Comments

Point 1: The title should reflect the effect (influence), (impact) of addition of Eu2O3 ceramic on the mechanical and microstructure of Ni-Cr alloy.

Response 1: Because the content of Eu2O3 is not given in Ni-20Cr-Eu2O3, it is conveyed that the content of Eu2O3 is changed, while the fear of increase’’ the impact of addition of Eu2O3 ceramic on the……’’ will lead to too lengthy title.

Point 2: English language needs moderate editing. Especially, from Abstract and Introduction sections. Sentence rearrangement is also necessary for better understanding.

Response 2: The English language has been moderate editing.

Point 3: Abstract section should be rewritten, just for example, line 19-21

Response 3: The abstract section has been rewritten.

Point 4: The manufacturer (provider) names for Ni, Cr and Eu2O3 powders are not specified in section 2.1.

Response 4: The manufacturer (provider) names for Ni, Cr and Eu2O3 powders has been specified.

Point 5: what does Bal. mean in Table 1?

Response 5: Bal. means that the remaining.

Point 6: EDX analysis described in line 154-157 should be added.

Response 6: Do you mean the description of the EDS test analysis? This part is described in the experimental testing section.

Point 7: -line 257: ... Eu2O3 ....Line 368

Response 7: The Eu2O3 has been revised

Point 8: Equations in the manuscript should be written by the mathematical identifier in Word (if the authors applied the manuscript using Microsoft Word). The terms in section 3.4 should be rewritten through mathematical identifier too.

Response 8: The equations in the manuscript has been rewritten.

Point 9: Based on the obtained outcomes of ultimate tensile strength, yield strength, elongation % and hardness, which composition is the optimum and will be preferred for the desired application?

Response 9: HP10 is the optimum and will be preferred for the desired application.

Round 2

Reviewer 1 Report

Line 24: Change ‘when’ to ‘with’ in the Abstrac.

Reviewer 2 Report

The authors have adequately addressed all the issues. Now I suggest the paper for publication in its present form.